# Spatiotemporal Variation Characteristics of Reference Evapotranspiration and Relative Moisture Index in Heilongjiang Investigated through Remote Sensing Tools

**Siyi Wen, Zihan Liu** [ID]**, Yu Han \*** [ID]**, Yuyan Chen, Liangsi Xu and Qiongsa Li**

College of Water Resources & Civil Engineering, China Agricultural University, 17 Tsinghua East Rd., Haidian District, Beijing 100083, China; wsy880880@163.com (S.W.); zihanliu1010@163.com (Z.L.); cyyvvv@cau.edu.cn (Y.C.); 2020309080110@cau.edu.cn (L.X.); sy20223092043@cau.edu.cn (Q.L.)
\* Correspondence: yhan@cau.edu.cn

**Abstract:** Reference evapotranspiration ($ET_0$) is one of the significant parameters in agricultural irrigation, especially in Heilongjiang, a big agricultural province in China. In this research, the spatiotemporal variation characteristics of evapotranspiration (ET), relative moisture index (MI) and influencing factors of $ET_0$ in Heilongjiang, which was divided into six ecology districts according to landforms, were analyzed with meteorological data observed over 40 years from 1980 and MOD16 products from 2000 to 2017 using Morlet wavelet analysis and partial correlation analysis. The results indicated that (1) the spatial distribution of ET and PET in Heilongjiang in humid, normal and arid years showed a distribution of being higher in the southwest and lower in the northwest, and higher in the south and lower in the north. The PET was higher than ET from 2002 to 2017, and the difference was small, indicating that the overall moisture in Heilongjiang was sufficient in these years. (2) In the last 40 years, the $ET_0$ increased while the annual MI decreased. The annual minimum of MI in the six regions of Heilongjiang was $-0.25$, showing that all six regions were drought free. (3) The importance of the meteorological factors affecting $ET_0$ was ranked as average relative humidity > average wind speed > sunshine duration. This research provides scientific guidance for the study of using remote sensing to reverse ET.

**Keywords:** evapotranspiration; relative moisture index; MODIS; remote sensing; Heilongjiang





## 1. Introduction

The grain production of China accounts for about a quarter of the total of the world, ranking first globally. The total grain output of Heilongjiang Province has ranked first in China for 13 consecutive years, indicating that Heilongjiang Province plays an important part in agriculture. Reference evapotranspiration ($ET_0$) is a vital index for the calculation of the water requirement of a crop and water resource evaluation in agriculture [1]. Therefore, $ET_0$ plays a key role in agricultural irrigation and maintaining a balance between water supply and demand [2]. Global climate change may cause a rise (or decline) of $ET_0$, thus, increasing (or decreasing) the likelihood of drought [3]. Hence, changes in $ET_0$ are closely related to drought conditions. Therefore, research on the spatiotemporal variation characteristics of evapotranspiration and drought conditions and their influencing factors in Heilongjiang would be of great help to provide a meaningful reference for the calculation of the water requirement of a crop and the optimal water management for agriculture production in this region.

With the increase in the rate of global warming [4], more and more scholars from all over the world have carried out research on the spatiotemporal characteristics of $ET_0$ from global to local scales. Dinpashoh et al. [5] proved that the $ET_0$ in Iran had a tendency to significantly increase first and then decline during 1965–2005. At the regional scale, the $ET_0$ in the Mediterranean region was reported to show an upward trend

by Vicente-Serrano et al. [6]. Patle et al. [7] discovered that the $ET_0$ in the eastern part of the Himalayas in Sikkim, India had a downward trend. Gan [8] demonstrated downward trends dominated in the $ET_0$ of Alberta, Canada. Xu et al. [9] explored the $ET_0$ trend in the Changjiang basin and pointed out that it showed a negative trend.

The drought situation in central and western China [3], North China [10] and the whole of northeast China [11,12] has been an important issue for domestic scholars. On the local scale, the drought conditions in Heilongjiang, which is a big agricultural province, are of particular concern. Based on the standardized precipitation index (SPI), Li et al. [13] revealed that the climate of Heilongjiang Province showed a trend of humidification on the whole, and the climate in spring, autumn and winter all had the same trend. Yan et al. [14] investigated the drought status in Heilongjiang Province on the basis of the MI and found that the MI in different periods in Heilongjiang Province had significant spatial differences. In previous research on drought in Heilongjiang Province, the directions of studies were mostly focused on the annual scale and seasonal scale of the whole province, and there has been a lack of analysis from the perspective of different landforms in Heilongjiang Province, as well as a lack of specific research at the monthly scale. In this research, Heilongjiang Province was divided into six regions according to various geographical characteristics and the $ET_0$ and MI were analyzed at the monthly scale.

Data on ET observed on the ground can present temporal variation, but it is hard to display the features of the spatial distribution of ET over a wide range, such as Heilongjiang Province in China, if there is not enough ground data. In contrast, remote sensing technology has been extensively used to study ET because it can obtain the surface conditions for a large area dynamically and quickly. In research by Dias et al. [15], the $ET_0$ in Brazil was roughly estimated using MOD16 data. Regionally, Castelli [16] analyzed MODIS data for the Alps and revealed that the $ET_0$ of forests and grasslands at higher altitudes exhibited positive trends, whereas negative trends prevailed at lower altitudes for grasslands and croplands in summer. Mu et al. [3] obtained the PET of midwestern China derived from MOD16 products and found that the general ET had a gradually increasing trend from the east to the southwest. Song et al. [17] based their research on ET in Xinjiang on MOD16 products and claimed that the distribution of the ET tendency rate in Xinjiang mainly showed a slight decreasing trend and the Hurst calculation showed that the climate in Xinjiang tended to be humid in the past 14 years. In addition, the accuracy of MOD16 products has been verified globally and locally. The ET data from MOD16 in North America was compared with the measured ET from 46 AmeriFlux vortex covariance flux towers in seven biological communities by Mu et al. [18]. The results indicated the accuracy of MOD16 had an average error of 24.1%, which is within the remote sensing error margin. Cheng et al. [19] reported that the MOD16 ET product showed an acceptable accuracy in China, with an average R-value of 0.87 and RMSE of 4.59 mm/8 d. Furthermore, Chang et al. [20] reported that the data from MOD16 products were in good agreement with measurements from the Tibet Plateau of China. However, few scholars have verified the accuracy of MOD16 in Heilongjiang. Therefore, the applicability of MOD16 in Heilongjiang will be verified in this research and the use of MOD16 products will be helpful to further study the spatiotemporal characteristics and changes in ET in Heilongjiang Province.

In this research, the spatiotemporal variation in PET and ET in Heilongjiang Province was analyzed using MOD16, the $ET_0$ in Heilongjiang Province was computed using the Penman–Monteith (P–M) formula with meteorological data, and changes in the $ET_0$ and relative moisture index (MI) were discussed. The influence factors of $ET_0$ in Heilongjiang Province were qualitatively studied using partial correlation analysis. The purpose of this research was to (1) analyze the spatiotemporal changes of PET and ET, (2) explore the spatiotemporal variation of $ET_0$ and MI and (3) study the characteristics of changing influencing factors of $ET_0$. This research is expected to offer a reference for the rational allocation and utilization of water resources in agriculture in Heilongjiang Province, China.

## 2. Materials and Methods

### 2.1. Study Area

Heilongjiang Province lies between longitudes 121.18°E and 135.08°E and latitudes 43.43°N and 53.55°N in northeastern China. Songnen Plain and Sanjiang Plain are important commodity grain bases in China. They are known as the 'great northern wilderness' and the main crops include corn, soybean and rice. Heilongjiang is a big agricultural province in China, so the adequacy of water resources is the main factor restricting agricultural development in this area. In the Chinese Ecosystem Assessment and Ecosystem Security Database (www.ecosystem.csdb.cn, accessed on 2 November 2022), ecological districts were classified according to the natural conditions such as ecosystem type and geographical characteristics. In Heilongjiang Province, these districts are the Sanjiang Plain ecoregion (I), Eastern Agriculture ecoregion (II), Greater Khingan Mountains ecoregion (III), Lesser Khingan Mountains ecoregion (IV), Western Meadow ecoregion (V) and Changbai Mountain ecoregion (VI). Thus, in an effort to further study regional features, Heilongjiang was divided into these six zones (Figure 1).

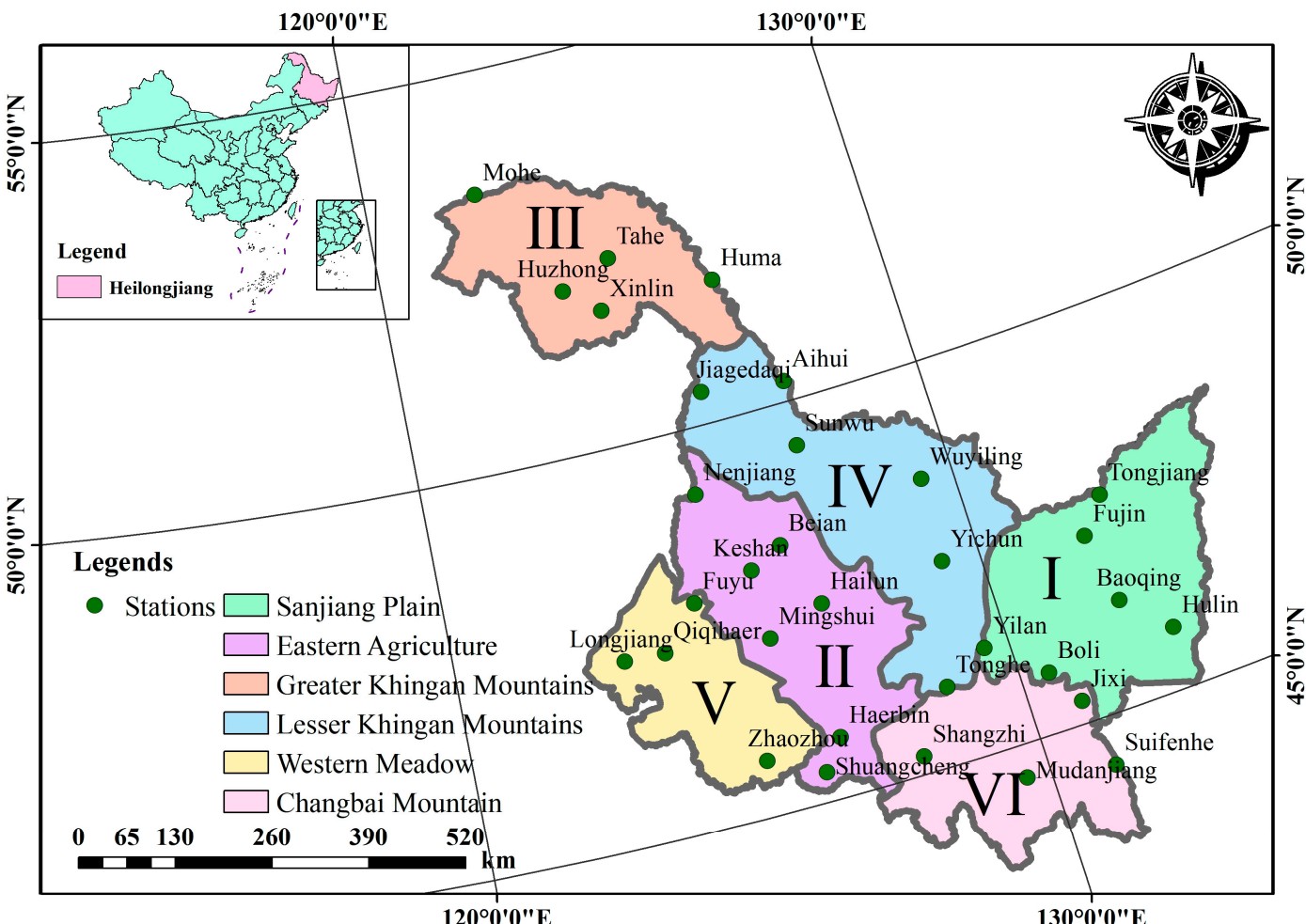

**Figure 1.** Distribution of meteorological stations and division of ecological districts in Heilongjiang. For the sake of presentation, there were some abbreviations in the later section. Region I symbolizes Sanjiang Plain (SP). Region II symbolizes Eastern Agriculture (EA). Region III is Greater Khingan Mountains (GKM). Region IV is Lesser Khingan Mountains (LKM). Region V is Western Meadow (WM). Region VI is Changbai Mountain (CM).

*2.2. Data Source*

2.2.1. MOD16 Products

The MOD16 A3 dataset includes surface ET, potential evapotranspiration (PET), latent heat flux (LE) and potential latent heat flux (PLE). The algorithm was developed on the basis of the P–M formula by Mu et al. [18]. The detailed information for MOD16 A3 is shown in Table 1. The downloaded product was the reprojected data from Heilongjiang Province in GeoTiff format under the WGS-1984/Geographic latitude and longitude coordinate system. According to MOD16 users' guide [21], the invalid values were removed and the valid values were multiplied by the coefficient to restore the true values in ARCGIS 10.8, and finally the PET of Heilongjiang province for 2000–2017 was obtained.

**Table 1.** Attributes of MOD16 products used in this study. They include product name, satellite orbit, time span, spatial resolution, temporal resolution and data source of MOD16 products.

| Product Name | Satellite Orbit | Time Span | Spatial Resolution | Temporal Resolution | Data Source |
|---|---|---|---|---|---|
| MOD16 A3 | h25v03/h25v04 h26v03/h26v04 h27v04 | January 2000–December 2017 | 500 × 500 m | 1 annum | USGS official website (https://lpdaac.usgs.gov/tools/appeears/, accessed on 16 January 2023) |

2.2.2. Meteorological Data

The meteorological data used in this paper were acquired from the National Meteorological Information Center (http://data.cma.cn/, accessed on 6 September 2022), including six meteorological parameters of average temperature, daily maximum temperature, daily minimum temperature, average relative humidity, average wind speed and sunshine duration from 32 meteorological stations (Figure 1) in Heilongjiang province from 1980 to 2019. In addition, daily evapotranspiration measured using evaporators was also obtained. The ET data from two meteorological stations (Wuyiling and Tongjiang) and ET data from 2018 to 2019 were missing. Thus, the spatiotemporal distribution of ET was analyzed without the data from the Wuyiling and Tongjiang meteorological stations, nor the data from 2018 to 2019.

*2.3. Calculation of Reference Evapotranspiration (ET$_0$) and Relative Moisture Index (MI)*

2.3.1. Reference Evapotranspiration (ET$_0$)

In this study, the Penman–Monteith (P–M) model suggested by the Food and Agriculture Organization (FAO), which has the highest international acceptance, was used to calculate ET$_0$ [22]. The formula is as follows:

$$ET_0 = \frac{0.408\Delta(R_n - G) + \frac{900\gamma}{T+273}U_2(e_s - e_a)}{\Delta + \gamma(1 + 0.34U_2)} \tag{1}$$

where $ET_0$ is the reference evapotranspiration (mm); $R_n$ is the net radiation at crop surface (MJ·m$^{-2}$d$^{-1}$); $G$ is the soil heat flux density (MJ·m$^{-2}$d$^{-1}$); $T$ is the average daily air temperature (°C); $U_2$ is the average daily wind speed at 2 m height (m·s$^{-1}$); $e_s$ is the saturation vapor pressure (kPa); $e_a$ is the actual vapor pressure (kPa); $e_s - e_a$ is the saturation difference (kPa); $\Delta$ is the slope of the vapor pressure curve (kPa·°C$^{-1}$); and $\gamma$ is the psychometric constant (kPa·°C$^{-1}$).

2.3.2. Relative Moisture Index (MI)

In the relevant drought research, in order to standardize and quantify it, a large number of drought indicators have been proposed by different scholars, such as standardized precipitation index (SPI) [23], Palmer drought severity index (PDSI) [24], reconnaissance drought index (RDI) [25], relative moisture index (MI) [26], etc. Among them, the relative moisture index, which takes into account the two modules of aerodynamic factors and energy balance, has good adaptability in the semi-humid area [27]. It was an ideal index with which to evaluate the drought situation in Heilongjiang. Therefore, the relative

moisture index (MI) was calculated to study the aridity in Heilongjiang Province in this research. According to the national standard, the calculation formula is as follows:

$$MI = \frac{P - ET_0}{ET_0} \times 100\% \tag{2}$$

where $P$ is the precipitation (mm), $ET_0$ is the total reference evapotranspiration (mm) and $MI$ is the relative moisture index.

With reference to the national drought class classification standard, the drought class of Heilongjiang Province was classified based on the relative moisture index (Table 2).

**Table 2.** Classification table of drought level according to the relative moisture index.

| Level | Type | Relative Moisture Index |
|:---:|:---:|:---:|
| 1 | No drought | $-0.40 < MI$ |
| 2 | Light drought | $-0.65 < MI \leq -0.40$ |
| 3 | Moderate drought | $-0.80 < MI \leq -0.65$ |

### 2.4. Validations of the MOD16 Data

To validate the MOD16 product in Heilongjiang, the ET data observed at the meteorological stations from 2000 to 2017 were used for correlation analysis with the PET data of the MOD16 product. Since PET indicates the maximum ET under unrestricted moisture conditions, and the measured ET data were also measured based on this condition, the applicability of MOD16 was verified by the correlation between the two. As shown in Figure 2a, the median and mean values of the two data sets were relatively close. The land surface ET showed two outliers in 2000 and 2001, which was consistent with the fact that a severe drought occurred in the spring and summer of northern China in 2000 and 2001. The similar box widths of ET and PET indicated that the volatility and stability of the two arrays were similar. In addition, by fitting the ET and PET data from 2000–2017, the correlation coefficient ($R^2$) [28,29] was found to be 0.1119 (Figure 2b). After excluding the two outliers from 2000 and 2001, the correlation coefficient ($R^2$) was reported to be 0.4621** (Figure 2b), which revealed that the positive trend of the PET from the MOD16 product and the ET observed at the meteorological stations was significant. The MOD16 data set could be applied to analyze the spatiotemporal distribution of ET in Heilongjiang.

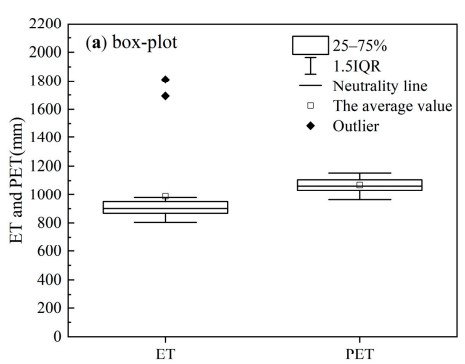
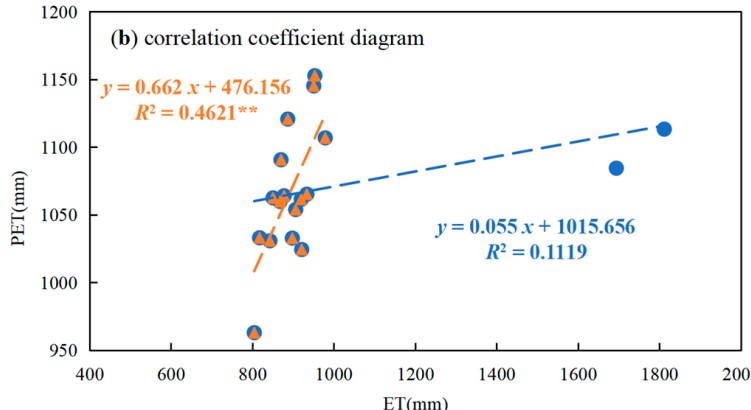

**Figure 2.** Comparison of ET and PET of Heilongjiang from 2000 to 2017: (**a**) box-plot; (**b**) correlation coefficient diagram. Blue circles represent years from 2000 to 2018. Orange triangles represent years from 2002 to 2018 (excluding two outliers). The increasing trend of the ET and PET from 2002 to 2017 passed a significant level test of 0.05. Note: ** indicates passing the significance level test of 0.05.

### 2.5. Research Methods

#### 2.5.1. Climate Tendency Rate

The climate tendency rate can reflect the variation trend of climate elements over a period of time. A one-way linear regression method [30] was adopted to analyze the temporal variation in the annual average ET in the study area, calculated as

$$S = \frac{n\sum_{j=1}^{n} j \times y_j - (\sum_{j=1}^{n} j)(\sum_{j=1}^{n} y_j)}{n \times \sum_{j=1}^{n} j^2 - (\sum_{j=1}^{n} j)^2} \tag{3}$$

where $n$ is the number of years, $y_j$ is the ET in the $j$th year and $S$ is the climate tendency rate. If $S > 0$, the ET increases during the $n$ years of monitoring and vice versa.

#### 2.5.2. Morlet Wavelet Analysis

Wavelet analysis [31] is widely applied to make the periodic analysis. It can not only reflect the periodic variation in a time series, but also help estimate qualitatively the future development trend. Therefore, in this manuscript, Morlet wavelet analysis was used to study the characteristic scales and periodicity of $ET_0$ and MI sequences. In Morlet wavelet analysis, signals or functions are basically represented or approximated by a system of cluster wavelet functions. The wavelet mother function is as follows:

$$\Psi(t) = e^{i\omega_0 t} e^{-\frac{t^2}{2}} \tag{4}$$

where $t$ is time (a), $\omega_0$ is a constant and $i$ is an imaginary unit.

The wavelet variance can be used for determining the relative strength of different scales of perturbations in the signal and the main cycle. The formula is as follows:

$$Var(a) = \int_{-\infty}^{+\infty} \left| W_f(a,b) \right|^2 db \tag{5}$$

where $W_f(a,b)$ is the wavelet coefficient.

#### 2.5.3. Partial Correlation Analysis

Partial correlation analysis [32] is a statistical method that, when two variables are correlated with a third variable at the same time, the effect of the third variable is removed and only the correlation coefficient of the other two variables is analyzed. This method takes into account and excludes the interactions between the elements, thus, reflecting the correlation accurately and qualitatively. In this manuscript, the partial correlation coefficients of each meteorological element and $ET_0$ were calculated to study the influencing factors of $ET_0$ in Heilongjiang.

It is assumed that variable Y is affected by variables $X_1$, $X_2$, $X_3$, ..., $X_n$ and $X_i$ (i = 1, ..., n), which are related to each other. Y and $X_1$ are represented linearly by the other variables $X_1$, $X_2$, $X_3$, ..., $X_n$:

$$Y = \widetilde{X}\theta_Y + e_Y \tag{6}$$

$$X_1 = \widetilde{X}\theta_{X_1} + e_{X_1} \tag{7}$$

where $\widetilde{X} = [X_1 \; X_2 \; ... \; X_n]$ is the regression variable vector, $\theta_Y$ and $\theta_{X_1}$ represent the parameter vectors, and $e_Y$ and $e_{X_1}$ are the estimated errors of the model.

## 3. Results

### 3.1. Spatiotemporal Distribution of ET and PET

The spatial distribution of ET and PET in Heilongjiang in the humid, normal and arid years during 2000–2017 was obtained using inverse distance weighting (Figures 3 and 4).

In the humid, normal and arid years, the spatial distribution of ET and PET varied greatly among the different regions in Heilongjiang Province, with the overall distribution being higher in southwest and lower in northwest, and higher in the south and lower in the north.

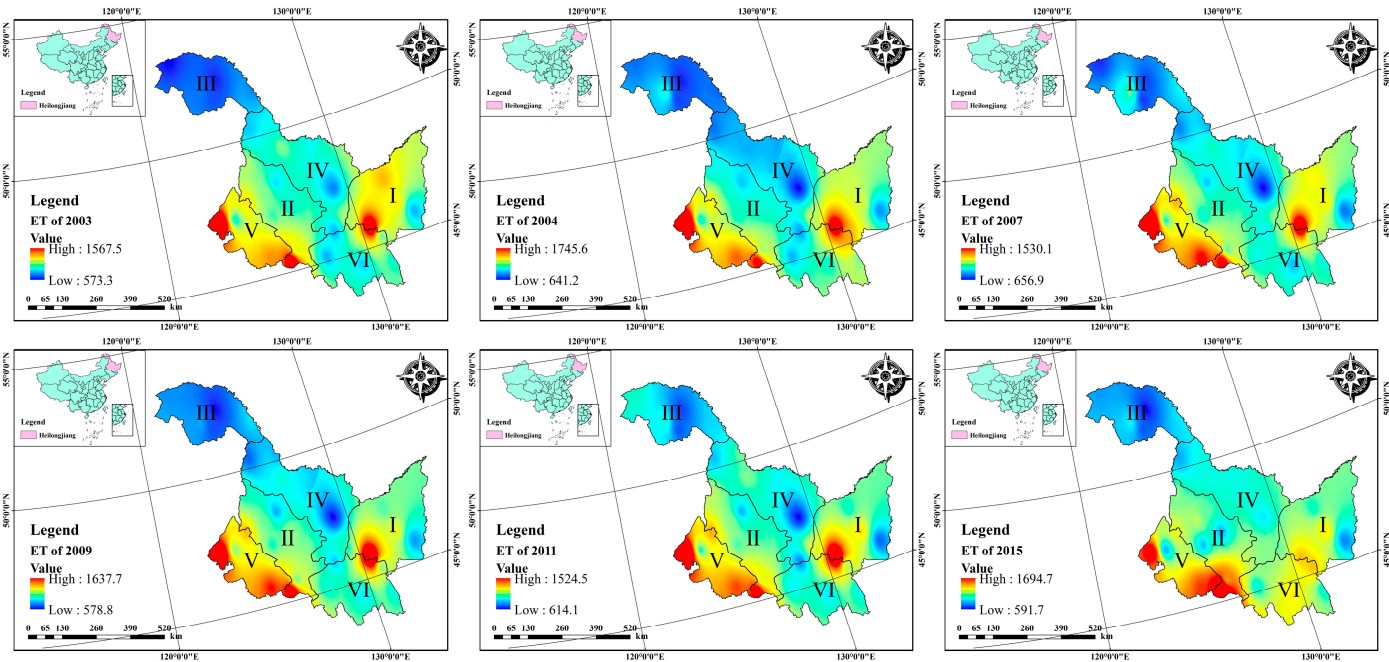

**Figure 3.** Spatiotemporal variation in ET in humid, normal and arid years from 2000 to 2017. 2003 and 2004 were humid years. 2009 and 2015 were normal years. 2007 and 2011 were arid years. Region I symbolizes Sanjiang Plain (SP). Region II symbolizes Eastern Agriculture (EA). Region III is Greater Khingan Mountains (GKM). Region IV is Lesser Khingan Mountains (LKM). Region V is Western Meadow (WM). Region VI is Changbai Mountain (CM).

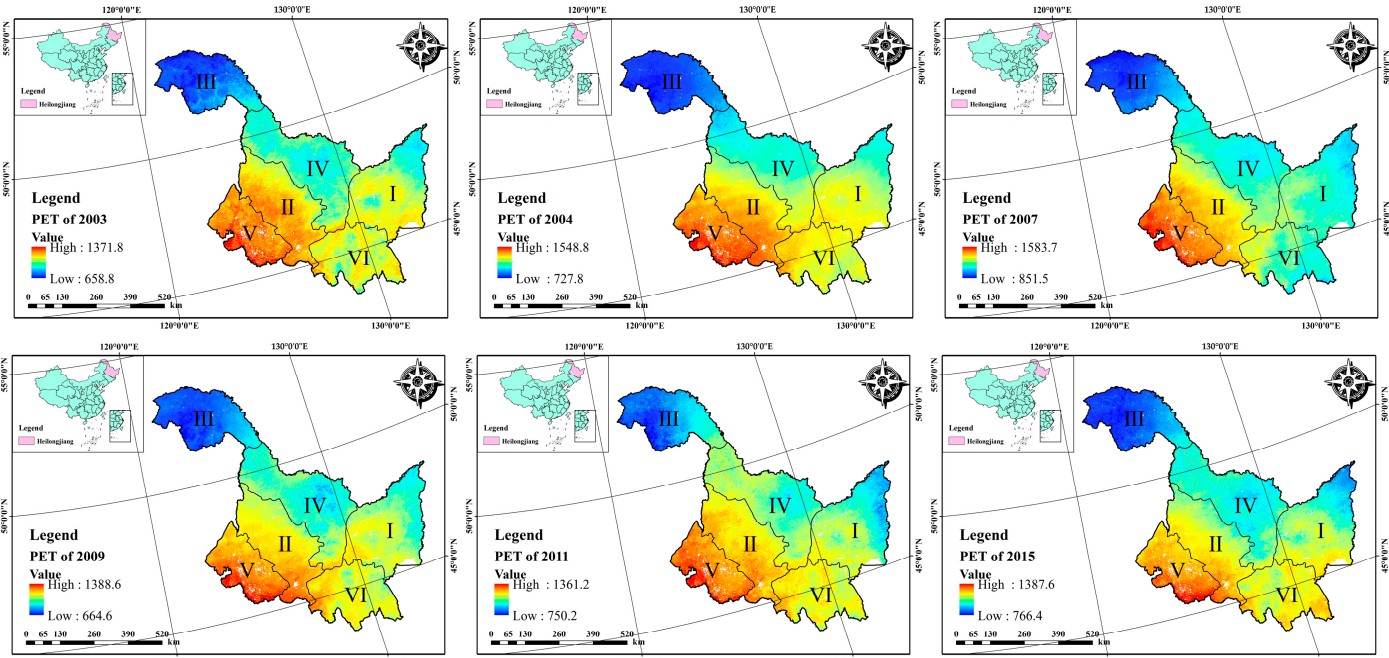

**Figure 4.** Spatiotemporal variation in PET in humid, normal and arid years from 2000 to 2017. 2003 and 2004 were humid years. 2009 and 2015 were normal years. 2007 and 2011 were arid years. Region I symbolizes Sanjiang Plain (SP). Region II symbolizes Eastern Agriculture (EA). Region III is Greater

Khingan Mountains (GKM). Region IV is Lesser Khingan Mountains (LKM). Region V is Western Meadow (WM). Region VI is Changbai Mountain (CM).

In the humid and arid years, the annual mean ET in Heilongjiang ranged from large to small in the order of WM > SP > EA > CM > LKM > GKM. ET in the normal year of 2009 had the same order, but ET in the normal year of 2015 was in the order of WM > EA > CM > SP > LKM > GKM, with the maximum value of 1060.33 mm for WM and the minimum value of 709.50 mm for GKM. It is worth mentioning that a high value center appeared at the junction of SP and CM in the humid, normal and arid years except the normal year of 2015. There was a low value center in LKM in the arid years, the normal year of 2009 and the humid year of 2004 (Figure 3).

As can be seen from Figure 4, the spatial distribution of PET in Heilongjiang Province changed little in the humid, normal and arid years. In Heilongjiang, the average PET of each region was in the order of WM > EA > CM > SP > LKM > GKM. The overall average value in Heilongjiang was 1070.36 mm and it decreased very slowly at a speed of 3.388 mm/a (Figure 5a). Compared with ET in Heilongjiang in the humid, normal and arid years, high value centers and low value centers in PET in Heilongjiang were less common than those in ET. Wang et al. [33] reported that the high ET area in Heilongjiang Province was located in the southwest, while the low ET area was located in the northwest, which was basically consistent with what was mentioned above.

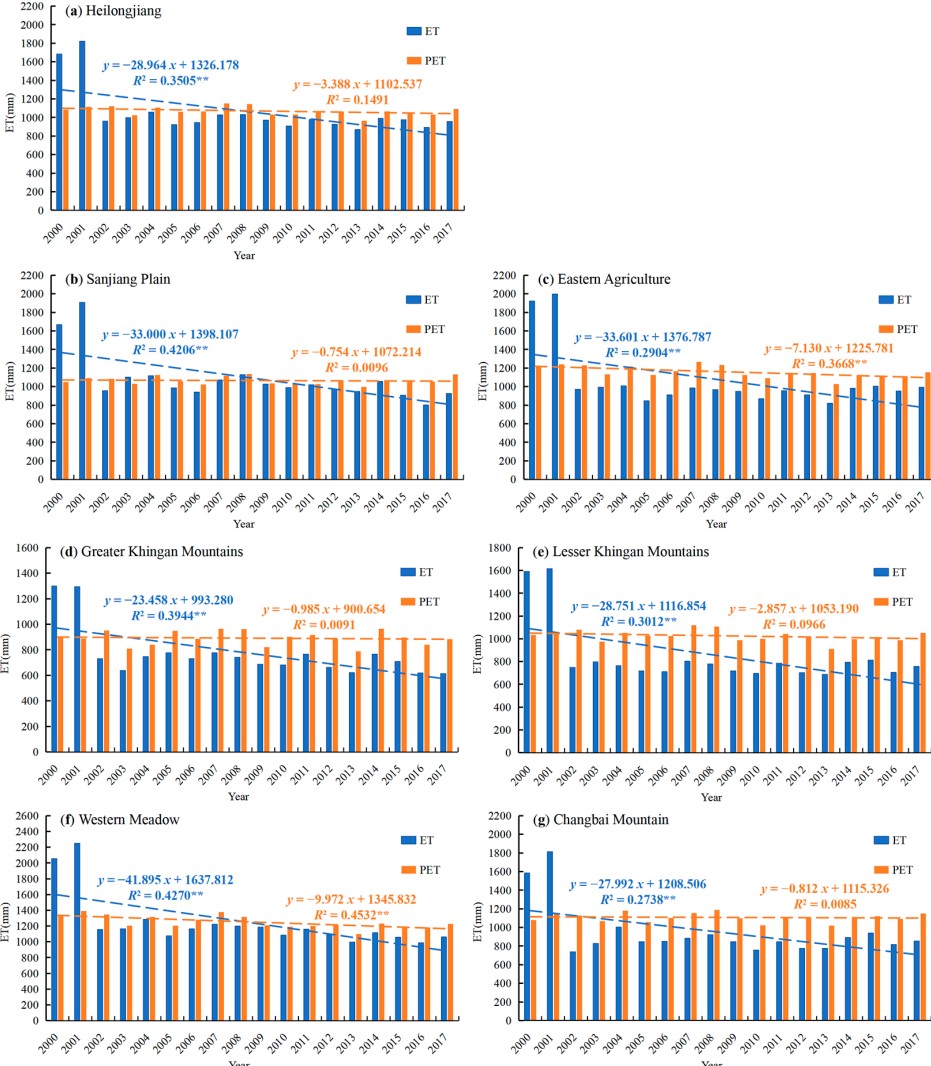

**Figure 5.** Interannual variation in ET and PET from 2000 to 2017 in Heilongjiang. Note: ** indicates passing the significance level test of 0.05.

ET in Heilongjiang in 2000 and 2001 was much higher than in other years (Figure 5), which was consistent with the conclusion of the ET outliers found above. From 2000 to 2017, the ET in all regions of Heilongjiang showed a significant decreasing trend, and the decreasing rate was in the order of WM > EA > SP > LKM > CM > GKM (Figure 5b–g), with the fastest decreasing rate of 41.895 mm/a in WM.

By contrast, the PET in Heilongjiang decreased very slowly at a speed of 3.388 mm/a (Figure 5). At the regional scale, the rate of decrease was in the order WM > EA > LKM > GKM > CM > SP (Figure 5b–g), with the fastest decreasing rate of 9.972 mm/a in WM. Except for the two drought years (2000 and 2001), the PET values were greater than ET values from 2002 to 2017, and the difference between the two was small, indicating that the water resource in Heilongjiang was sufficient in these years and there was no problem of drought and water shortage.

### 3.2. Interannual Variation in $ET_0$, Precipitation and MI

The linear regression analysis suggested (Figure 6) that, from 1980 to 2019, the average annual precipitation in the study area showed a significant decreasing trend at a rate of 3.707 mm/a. In contrast, the average annual $ET_0$ increased at a rate of 0.002 mm/a from 1980 to 2019. It should be noted that the change in annual mean precipitation was statistically significant, while the change in annual mean $ET_0$ was not statistically significant.

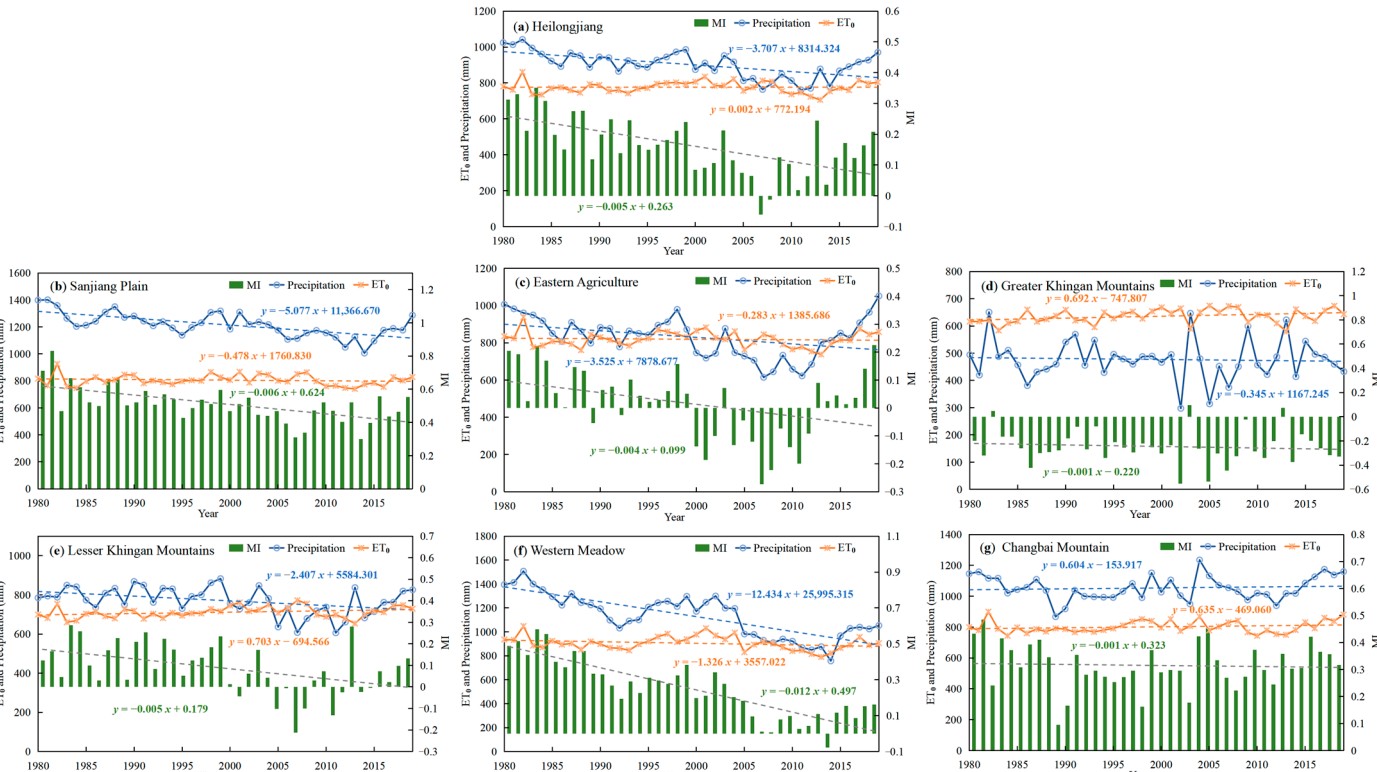

**Figure 6.** Interannual variation in $ET_0$, precipitation and MI in Heilongjiang during 1980–2019.

Among the six regions in Heilongjiang, the annual mean $ET_0$ had a positive trend in GKM, LKM and CM, with LKM increasing at the fastest rate of 0.703 mm/a. Meanwhile, the decreasing trends in the other regions were observed from Figure 6, with WM showing the fastest decrease at a rate of 1.326 mm/a. The average annual precipitation in all zones except CM followed a decreasing trend, with the decreasing rate from largest to smallest being WM > SP > EA > LKM > GKM, and the maximum in WM was as high as 12.434 mm/a.

So as to better evaluate the moisture level and crop growth conditions in Heilongjiang Province, the relative moisture index (MI) of each region in Heilongjiang Province was

calculated. The average annual MI from 1980 to 2019 in Heilongjiang Province was found to decrease at a rate of 0.005/a. It should be noted that the change in annual MI was statistically significant.

The negative trends of the annual mean MI occurred in the six ecological districts, and the decreasing rate was in the order of WM > SP > LKM > EA > GKM > CM, with the maximum of 0.012/a in the WM. From 1980 to 2019, the average annual MI of the six ecological regions in descending order was SP > CM > WM > LKM > EA > GKM, with a minimum value of −0.25. Furthermore, the average annual MI in Heilongjiang was reported as 0.16. According to the classification table of drought levels (Table 1), Heilongjiang and its six ecological regions were all without drought, indicating that the study area had an abundant water supply all year round, which could meet the nourishing conditions of grain crops in Heilongjiang.

### 3.3. Monthly Variation in $ET_0$, Precipitation and MI

The monthly mean $ET_0$ and precipitation in Heilongjiang showed significant seasonal variations (Figure 7). In general, both $ET_0$ and precipitation increased first and then decreased on the monthly scale. Although the study area was rainy in summer and less rainy in winter, it was more humid in winter than in summer because the $ET_0$ in summer was greater than precipitation and the $ET_0$ in winter was lower than precipitation. In June, the $ET_0$ reached its maximum value of 131.53 mm, while the peak value of precipitation was 97.93 mm in July.

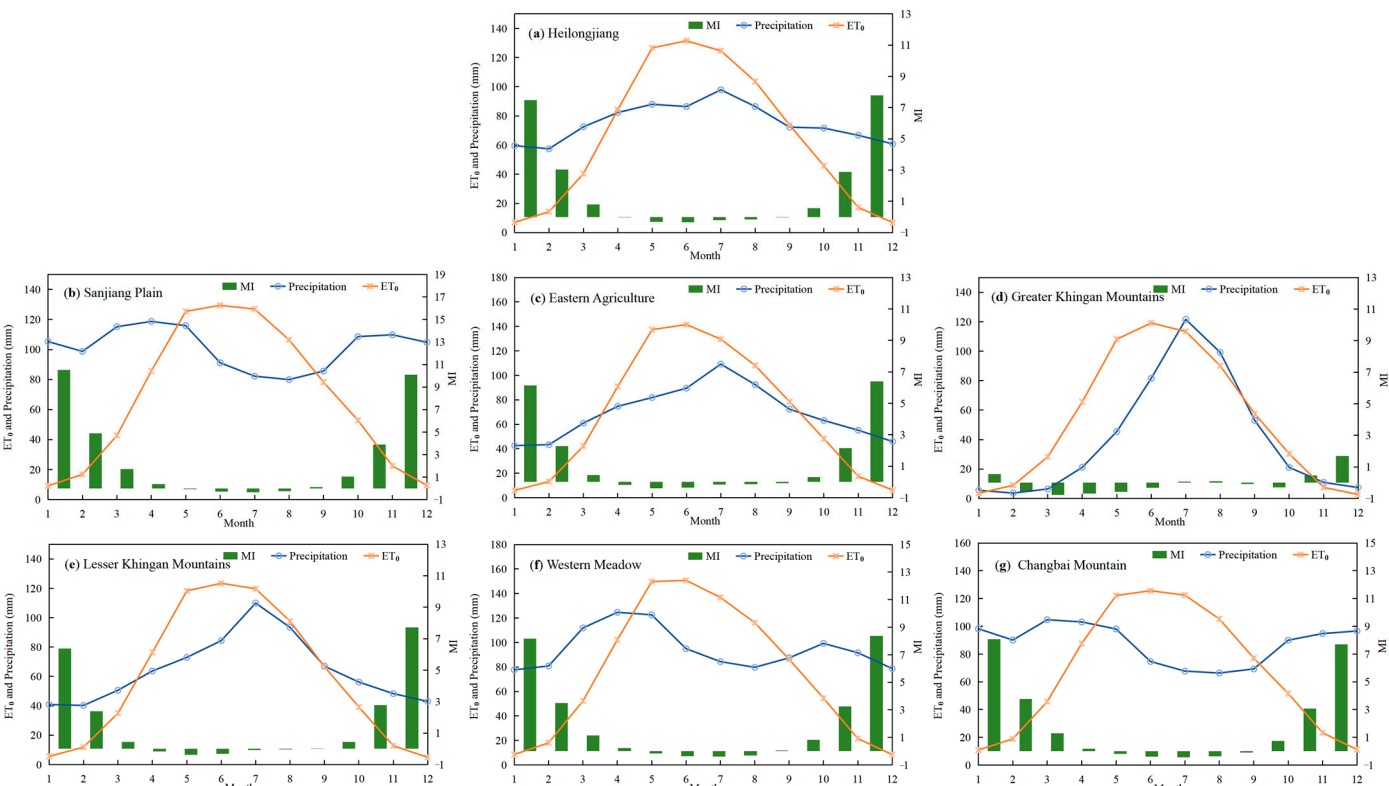

**Figure 7.** Average monthly variation in $ET_0$, precipitation and MI in Heilongjiang.

The monthly average $ET_0$ of each ecological region was higher in summer and lower in winter. The average monthly precipitation in EA, GKM and LKM had the same features as the $ET_0$. However, in other regions, the average monthly precipitation from January to December was revealed as a special trend of first increasing, then decreasing and finally increasing.

The month with maximum $ET_0$ in the six ecological zones was June, and the maximum $ET_0$ of the SP, EA, GKM, LKM, WM and CM regions was 129.44, 141.49, 119.15, 123.42,

150.76 and 125.62 mm, respectively. The lowest $ET_0$ in SP, EA and CM was 9.14, 5.96 and 10.83 mm in January. The lowest monthly $ET_0$ of GKM, LKM and WM was 2.75, 4.93 and 8.40 mm, respectively, which all occurred in December. In addition, the maximum precipitation of SP and West Meadow occurred in April, which was 118.70 and 124.64 mm, respectively. Meanwhile, the maximum monthly precipitation in CM was 104.71 mm in March. The maximum monthly precipitation in EA, GKM and LKM was 109.20, 121.58 and 109.96 mm, respectively, all recorded in July. The lowest precipitation in SP and CM was in August, with values of 79.96 and 66.32 mm, while the minimum in the GKM and LKM regions was 3.68 and 40.26 mm in February. The lowest monthly precipitation in EA and WM was 42.61 and 77.67 mm, respectively, both of which occurred in January.

It could be easily seen from Figure 7 that the average monthly MI of Heilongjiang Province was also seasonally variable. However, the monthly MI exhibited the opposite trend to the monthly $ET_0$ and precipitation, as it first decreased and then increased from January to December, and was greater than −0.34. Both the temporal trends of the $ET_0$ and precipitation resulted in the lowest MI of −0.34 in June. It is worth noting that the MI was above 0 from January to March and from October to December.

From the perspective of the six ecological districts, the monthly mean MI in the GKM region had the smallest variation, while it varied significantly in other zones, where MI was low in summer and high in winter.

The maximum MI of SP and CM occurred in January, with values of 10.51 and 8.07, respectively. The highest MI values were primarily found in the EA, GKM, LKM and WM in December, with 6.40, 1.70, 7.72 and 8.37, respectively. Meanwhile, the lowest MI in EA and LKM was −0.40 and −0.38, respectively, and these were recorded in May. The lowest MI in the GKM was reported as −0.77 in March. The lowest monthly MI in SP, West Meadow and CM were −0.35, −0.38 and −0.45, respectively, all of which appeared in July. Generally, Heilongjiang was wet without drought on the whole, but there was drought in some parts in some months. According to the classification table of drought levels (Table 1), there was light drought in the EA region in May, light drought in the CM region in June and July, light drought in the GKM region in February and May, and moderate drought in the same region in March and April.

*3.4. Periodic Characteristics of $ET_0$ and MI*

Providing the purpose of exploring the change period of $ET_0$ and MI in Heilongjiang Province, Morlet wavelet analysis was adopted to explore the scale and oscillation characteristics of $ET_0$ and MI during 1980–2019. It could reflect the features of time domain and frequency domain of $ET_0$ and MI.

### 3.4.1. Periodic Characteristics of $ET_0$

Wavelet analysis (Figure 8a) showed that the $ET_0$ was clearly localized on an annual time scale, with uneven time and frequency domain distribution. The maximum peak corresponded to a 28-year time scale, indicating that periodic oscillations were strongest at about 28 years. Therefore, 28 a is the first major period of the annual $ET_0$. The second peak corresponded to the time scale of 5–7 a, and 6 a was the second major period (Figure 8b). The positive and negative phases of the annual $ET_0$ wavelet coefficients changed alternately on the time scales of 6 a and 28 a, respectively. The mutation points of the wavelet coefficients appeared about every 2 years and 8–9 years (Figure 8c), that is, the annual trend changed once.

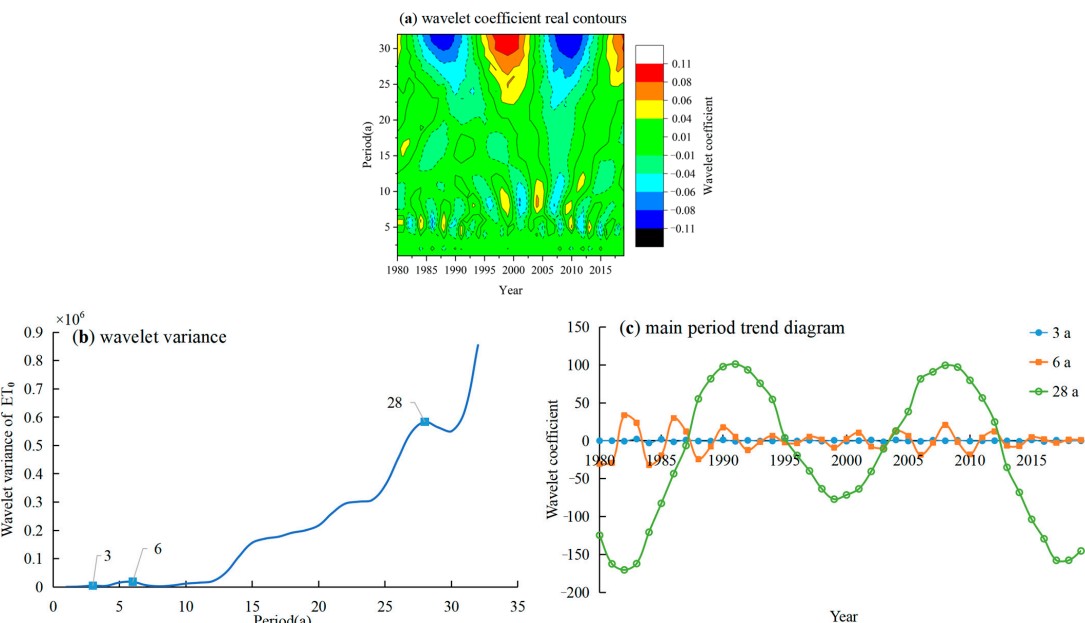

**Figure 8.** Morlet wavelet analysis of periodic characteristics of average $ET_0$ in Heilongjiang from 1980 to 2019: (**a**) wavelet coefficient real contours, (**b**) wavelet variance and (**c**) main period trend'diagram.

### 3.4.2. Periodic Characteristics of MI

It was revealed from Figure 9a that MI, like $ET_0$, also had significant localization characteristics on the interannual time scale. The annual MI varied with a medium period of 16–18 a and a short period of 4–9 a. As shown in Figure 9b, there were three obvious peaks, corresponding to the time scales of 5, 8 and 17 a, which indicated that annual MI had the first major period of 5 a, the second major period of 8 a and the third major period of 17 a. The positive and negative phases of the annual MI wavelet coefficients varied alternately on the time scales of 5, 8 and 17 a, respectively. The abrupt points of the wavelet coefficients arose about every 3–4 years, 5–6 years and 10–11 years (Figure 9c), that is, the annual MI trend changed once.

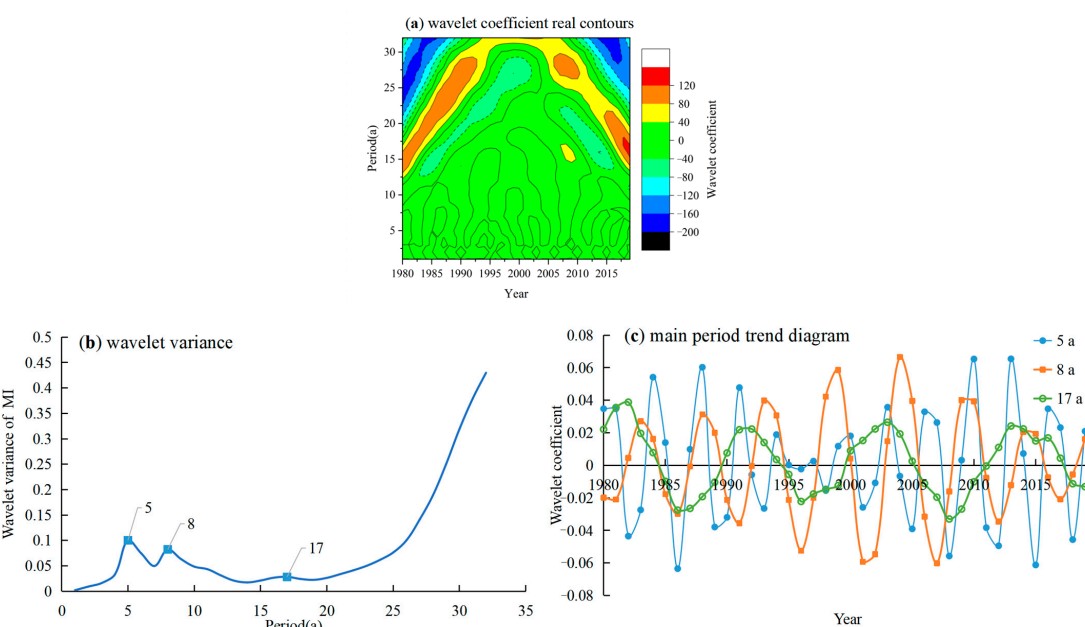

**Figure 9.** Morlet wavelet analysis of periodic features of average MI in Heilongjiang from 1980 to 2019: (**a**) wavelet coefficient real contours, (**b**) wavelet variance and (**c**) main period trend diagram.

### 3.5. Influencing Factors of $ET_0$ and Their Trend Characteristic

Six meteorological factors that influenced $ET_0$ during 1980–2019, including average temperature, daily maximum temperature, daily minimum temperature, average relative humidity, average wind speed and sunshine duration, were analyzed using partial correlation analysis, and the partial correlation coefficients between the annual mean meteorological factors and the annual mean $ET_0$ were calculated. Figure 10 illustrated that the main factors affecting $ET_0$ in Heilongjiang were average wind speed, sunshine duration and average relative humidity. To be specific, the main factors affecting the GKM region were average wind speed and sunshine duration. The factors that affected the LKM region were sunshine duration and average relative humidity. In SP and EA, the $ET_0$ was mainly influenced by sunshine duration, average wind speed and daily minimum temperature. In WM, the average temperature, daily minimum temperature, average relative humidity and average wind speed played a major role in affecting the $ET_0$. Furthermore, average relative humidity, sunshine duration and average wind speed influenced the $ET_0$ in CM. Moreover, the factors that were positively correlated were average temperature, sunshine duration and average wind speed, while the average relative humidity and daily minimum temperature were negatively correlated with $ET_0$. The higher daily maximum temperature was, the lower the $ET_0$ was in GKM, EA and WM. In contrast, it was positively correlated in LKM, SP and CM.

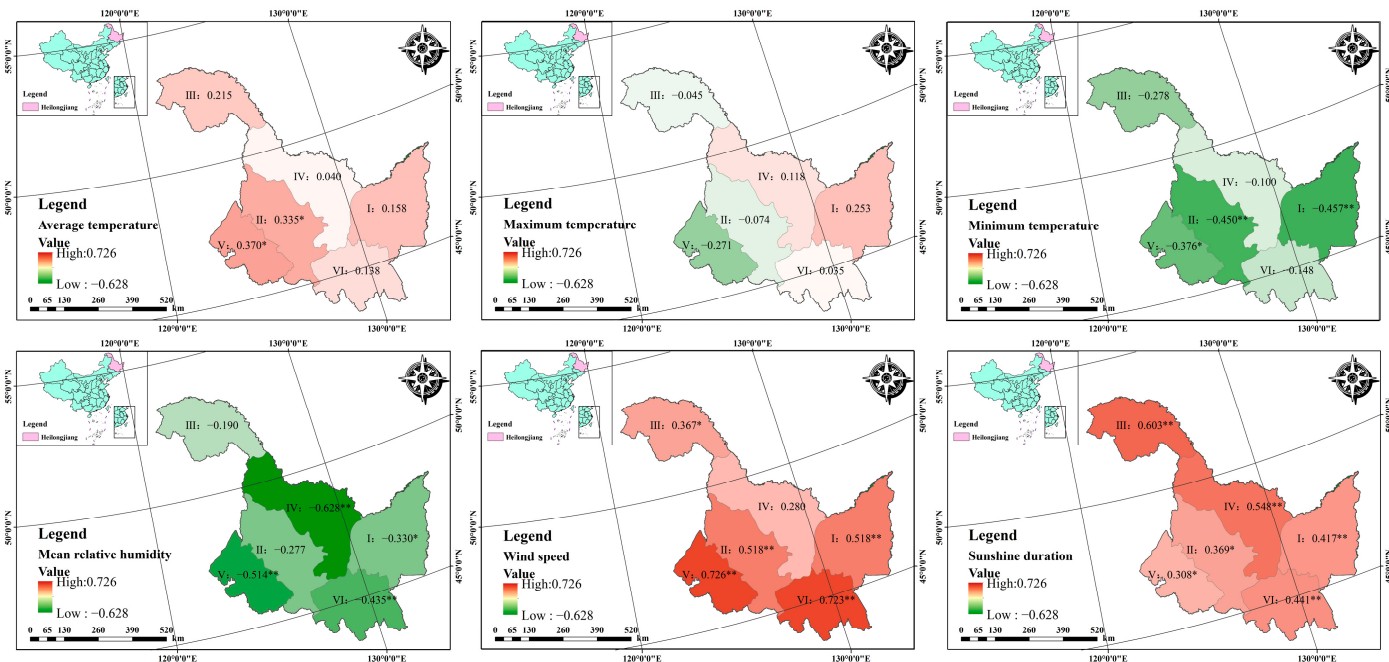

**Figure 10.** Partial correlation coefficient of influencing factors of $ET_0$ in Heilongjiang Province from 1980 to 2019. Note: * and ** indicate passing the significance level test of 0.1 and 0.05, respectively.

The variation trend of influencing factors in Heilongjiang was further analyzed as follows. Figure 11 suggested that from 1980 to 2019 the average temperature, daily maximum temperature and daily minimum temperature in Heilongjiang significantly increased, while the average relative humidity showed a downward trend. Specifically, the average relative humidity exhibited a significant increasing trend in SP, but decreased in other ecological districts. The average relative humidity in GKM and LKM decreased rapidly with a tendency rate of −0.074 and −0.081%/a, respectively. In Heilongjiang, it was known that the average wind speed had a downward trend. Considering the six ecological districts, the tendency rate of CM increased slowly by 0.002 m/s/a, but decreased in other regions, and decreased fastest in WM at 0.034 m/s/a. The sunshine duration decreased in Heilongjiang Province on the whole, but increased in GKM and LKM, while it decreased in

other regions. The sunshine duration decreased remarkably in EA with a tendency rate of 0.012 h/a, but did not change significantly in other regions. Furthermore, the overall change in sunshine duration and average wind speed in Heilongjiang Province was small, while the variation rate of average relative humidity was large. As a result, the increase in the $ET_0$ in Heilongjiang Province during 1980–2019 was mainly attributed to the variation in average relative humidity. Wang et al. [31] found that relative humidity was the main factor affecting the change in evapotranspiration of reference crops in Heilongjiang on an annual scale, which was consistent with the above conclusion.

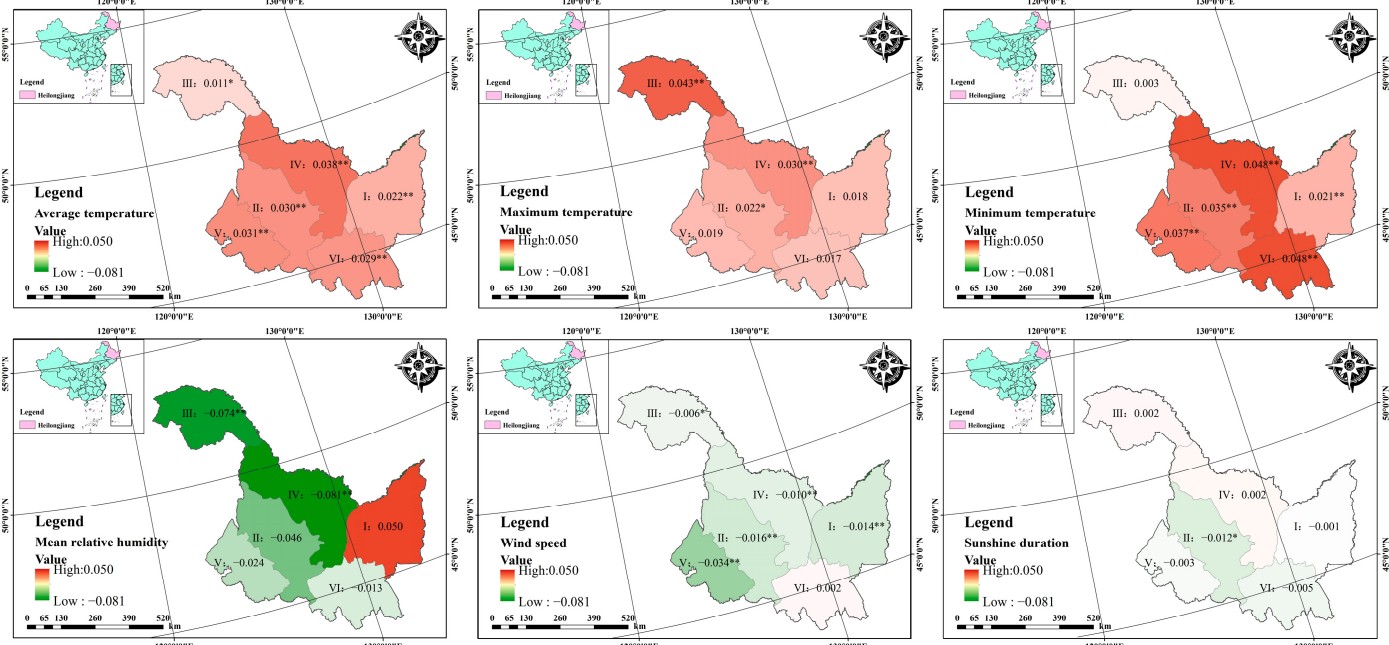

**Figure 11.** Variation trend of influencing factors of $ET_0$ in Heilongjiang Province from 1980 to 2019. Note: * and ** indicate passing the significance level test of 0.1 and 0.05, respectively.

## 4. Discussion

### 4.1. Applicability of the MOD16 Products in Humid and Semi-Humid Areas

Recently, MOD16 products have been widely used in hydrological research about the $ET_0$ of humid and semi-humid areas. Therefore, it is important and worthwhile to explore the applicability of the MOD16 products in humid and semi-humid areas. In this study, the MOD16 products in Heilongjiang were validated. The median and mean values of the PET from the MOD16 products and the ET from the meteorological station were close when the outliers were excluded. Additionally, the correlation coefficient ($R^2$) was found to be 0.4621**. It did not show a good linear relationship, but the increasing trend of the PET and the ET was significant, indicating that the MOD16 products could apply to Heilongjiang. It was found that there was an error in the MOD16 PET data compared with the measured ET in two drought years, 2000 and 2001 and the ET values were nearly twice as high as the PET values in these two years. Thus, it was possible that when major regional climate changes occurred, the MOD16 algorithm was unable to monitor them and make corresponding changes. Scholars at home and abroad have performed a lot of verification on MOD16 products in other humid and semi-humid areas. He [34] compared the grid value of daily ET from MOD16 products in Jizhou District of Tianjin with the corresponding grid value of local observed ET, and found that the absolute error was less than 1.00 mm/d, and the average relative error was 11.63%. When Andrade et al. [35] estimated LE in order to verify the accuracy of MOD16 data in the Amazon forest, they revealed that the annual mean value obtained using the eddy covariance system verified a good approximation of the model. Chen et al. [36] analyzed the difference between the data from MOD16 products

in the middle and lower Yangtze Plain and PET values calculated using the P–M formula, and claimed that the $R^2$, mean relative error and mean root error were 0.67, 10.40% and 0.31 mm/d, respectively, indicating a good match between them. The experimental ET from flooded rice fields was compared with the MOD16 ET products in southern Brazil by Souza et al. [37] and it was discovered that the algorithm could not detect areas with high soil moisture. Du et al. [38] assessed the validation of the MOD16 products of Panjin coastal wetland with flux tower accumulation ET data, and presented the result that the deviation in summer was 6.14 mm/8 d, suggesting that the accuracy of MOD16 ET products was low. Overall, the results of the validation studies for MOD16 products varied from zone to zone in humid and semi-humid areas. The inconsistencies between MOD16 ET and measured ET may be caused by the parameterization of the P–M model, flux tower measurement errors, MODIS pixels and limitations of the algorithm used by MOD16, etc. [38]. All in all, more research is needed to further verify the accuracy of MOD16 in humid and semi-humid areas and reduce the uncertainty of MOD16 products, so as to better explore the variation in ET. In addition, the short duration of the MOD16 dataset, since the satellite named Terra with the MODIS sensor was launched in late 1999, limits the time scale for studying ET changes. Therefore, if scholars want to study ET variation over a larger time scale, it is better to introduce data from other satellites.

*4.2. Cause Analysis of the Variation Trend of the Influencing Factors of $ET_0$*

From 1980 to 2019, the mean temperature, daily maximum temperature and daily minimum temperature in Heilongjiang all exhibited a positive trend, while the mean relative humidity, average wind speed and sunshine duration showed the opposite trend. The cause analysis of the variation trend of each influence factor of $ET_0$ showed that the main reason for the rising trend of average temperature, daily maximum temperature and daily minimum temperature was the continuous accumulation of greenhouse effects, which have led to global warming [39]. The negative trend of precipitation in Heilongjiang from 1980 to 2019 was probably responsible for the decrease in its average relative humidity. The fact of decreasing wind speed in Heilongjiang was mainly caused by the decrease in wetland area, the increase in cultivated land area and global warming [40]. The weakening of sunshine duration in Heilongjiang may be closely related to urbanization. Due to urbanization, anthropogenic aerosols and air pollutants emissions rapidly increased, increasing the amount of cloud cover and, hence, resulting in declining sunshine duration. Meanwhile, the large-scale monsoon circulation has weakened over recent decades, which led to a decrease in wind speed, probably bringing about the downward trend in sunshine duration in Heilongjiang [41]. Additionally, the interannual variation trend of various meteorological factors could also be affected by the number of observation sites and the time range studied.

## 5. Conclusions

Based on the meteorological data of 32 meteorological stations in Heilongjiang from 1980 to 2019 and MOD16 PET data from 2000 to 2017, the spatiotemporal variation characteristics of ET and MI in Heilongjiang, which was divided into six ecological districts, were primarily analyzed. Meanwhile, the variation features of the influencing factors of the $ET_0$ were studied using partial correlation analysis. The following conclusions were reached:

(1)     After removing outliers from ET and PET in Heilongjiang from 2000 to 2017, it was found that the volatility and stability were similar, and the correlation coefficient ($R^2$) reached 0.4621**, indicating a significantly increasing trend. The spatial distribution of ET and PET in the humid, normal and arid years varied greatly among different regions in Heilongjiang, showing a general distribution pattern of being higher in the southwest and lower in the northwest, and higher in the south and lower in the north. Compared with ET in Heilongjiang in the humid, normal and arid years, high value centers and low value centers in PET in Heilongjiang were less common than those in ET. Except for two drought years, 2000 and 2001, PET was greater than ET

in Heilongjiang from 2002 to 2017, and the difference between the two was small, indicating that the overall moisture in Heilongjiang was sufficient in these years.

(2)　From 1980 to 2019, the average annual precipitation in the study area decreased significantly at a rate of 3.707 mm/a, the average annual $ET_0$ increased at a rate of 0.002 mm/a and the average annual MI decreased at a rate of 0.005/a. At the annual scale, Heilongjiang and its six ecological regions were drought free. At the monthly scale, monthly $ET_0$ and monthly precipitation in Heilongjiang increased first and then decreased, while monthly MI showed the opposite trend to monthly $ET_0$ and monthly precipitation. Heilongjiang was generally wet, but there was drought in some parts in some months. The EA district experienced light drought in May, the CM district experienced light drought in June and July, and the GKM experienced it in February and May, as well as moderate drought in March and April. The first primary period of annual $ET_0$ was 28 a and the second primary period was 6 a. The interannual MI had three main periodicities: 5, 8 and 17 a. This research filled the research gap of studies on $ET_0$ at a monthly scale in various ecological regions of Heilongjiang Province.

(3)　The main factors affecting $ET_0$ in Heilongjiang Province from 1980 to 2019 were average wind speed, sunshine duration and average relative humidity. On a 40-year scale, the average temperature, daily maximum temperature and daily minimum temperature in Heilongjiang all exhibited a positive trend to varying degrees, while the average relative humidity, average wind speed and sunshine duration had a decreasing trend. Since the sunshine duration and average wind speed in Heilongjiang Province exhibited few changes, while the average relative humidity exhibited a large change rate, it was believed that the increase in $ET_0$ in Heilongjiang Province during 1980–2019 was mainly caused by the variation in average relative humidity.

This research is expected to provide a reference for the calculation of water requirements of crops and the optimal water management for agriculture production in Heilongjiang Province, China.

**Author Contributions:** Conceptualization, S.W.; methodology, S.W.; validation, S.W.; formal analysis, S.W.; investigation, S.W., Z.L., Y.H., Y.C., L.X. and Q.L.; resources, S.W., Z.L., Y.H., Y.C., L.X. and Q.L.; data curation, S.W. and Y.H.; writing—original draft preparation, S.W.; writing—review and editing, S.W.; visualization, S.W.; project administration, Y.H.; funding acquisition, Y.H. All authors have read and agreed to the published version of the manuscript.

**Funding:** This research was funded by the National Natural Science Foundation of China under grant number 51979275; the 2022 Science and Technology Think Tank Program under grant number 20220615ZZ07110111; the Joint Foundation of Tarim University and China Agricultural University under grant number ZNLH202205; and the Joint Open Research Fund Program of the State key Laboratory of Hydroscience and Engineering and Tsinghua–Ningxia Yinchuan Joint Institute of Internet of Waters on Digital Water Governance under grant number sklhse-2022-Iow08.

**Data Availability Statement:** The data used in this study are available upon request from the first author.

**Conflicts of Interest:** The authors declare no conflict of interest.

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
