# Peer review of "Spatiotemporal Variation Characteristics of Reference Evapotranspiration and Relative Moisture Index in Heilongjiang Investigated through Remote Sensing Tools"

_remotesensing, doi:10.3390/rs15102582_

Round 1
Reviewer 1 Report
In this paper, the author analyzed the spatiotemporal characteristics of ET and MI in heilongjiang province, as well as the influencing factors of ET0, using long-term observational meteorological data and MOD16 products. It is an interesting work. While, there are many problems that need expatiation. I have provided the following comments to help improve the quality of the paper.
1 In introduction, the author should specify the originality and main contributions of this article.
2 The logic of the literature review is confusing. The introduction section should clarify the research motivation and explain clearly the relationship between ET0 and drought.
3 In manuscripts, the use of first-person should be avoided as much as possible, except for certain special cases. Therefore, please delete all first-person words from the manuscript according to the instructions.
4 There is a lack of detailed information about the modis products used, and there is no explanation about why the relative moisture index was used. Relevant explanations regarding the selection of the relative moisture index are missing in the text.
5 Lines57-58, The evapotranspiration observed on the ground is often not accurate enough due to the limitations of natural and man-made conditions, the description is not rigorous enough, please check and modify it.
6 Lines476-477, “the correlation coefficient (R2) reached 0.4621, indicating a good correlation.” What are the author's criteria for evaluation? It's not clear for me.
7 Figure 1, above The Sanjiang Plain ecoregion (I), the line is not clear enough. Please make corrections.
There are small spelling errors in the text, for example in line 45. please review the entire text and correct them.
Reviewer 2 Report
The manuscript entitled “Remote sensing based research…” analysis the climatic conditions related to water availability at ground level (precipitation, evaporation) in Heilongjiang region of China. The study presents interest for the readers by its vast analyzed region located in cold climatic conditions, but also by the complex methodological approach on which various statistical methods are deployed. Despite these general strong points, the study presents some major weaknesses that should lead to a thorough revision of the manuscript if they will be properly addressed. Among these I can mention the elementary description of the results, the poor synthesis, the fragile discussion of the results and the visible flaw in the interpretation of the obtained results. My comments are listed below and I hope to be useful in the rethinking of the manuscript.
Major comment:
Figure 5 shows confusing colors for regional trend lines. The orange like trend line is used in the graph on the top for PET, but for regional trend only ET is shown. Also, the equation on the graphs should be completed with R-squared values of these trend lines which can sustain statistical significant increasing or decreasing trends. However, the decreasing trend line shown is not consistent. Actually, all we can see are two years with high evaporation at the beginning of the study period followed by largely constant values. This elementary aspect raises my major concern regarding the scientific soundness of the entire methodological approach.
Specific comments:
The title seems as a report. I suggest: ”Spatio temporal variation characteristics of reference evapotranspiration and relative moisture index in Heilongjiang investigated through remote sensing tools”
The abstract is a little bit incoherent and does not highlight the novelty of the study at local/regional scale.
The Introduction is poorly structured and the main ideas of each paragraph are impossible to be tracked by the reader. I recommend a better organization of the delivered information that could be arranged from global to local. Also, here we have a regional study, thus the authors should reassess the relevance of some cited papers dealing with the topic of the manuscript, but in remote regions of the world.
In section 2.5 the explanation is too didactical and general. Actually, this part should not insist so much on the explanations of the methods, since they are not new on the field, but on how they responded to the study scientific goals and demands.
The distribution and the analysis should be presented in a geographical manner (e.g. plain region higher evaporation than in mountain area). Comparing regions is hardly understandable by the common international reader (e.g. last paragraph of page 9).
It does not emerge clearly the criteria for the grouped years shown in Figure 3 and 4. I suggest to synthetize this by clustering the years in two large categories based on the regional precipitation amount: humid and arid years.
Elementary description with weak scientific consistency is often present in the text (as in Paragraph 1 of subsection 3.3).
Subsection 4.1 is more fit with Introduction section, because actually does not discuss the manuscript results.
Subsection 4.2 indicates that cloud cover and aerosols are decreasing and sunshine is also decreasing. This is not coherent.
Minor comments:
L12: Please give the full name for ET at the first use, even if we can suppose that it is evapotranspiration
L17: higher instead of greater
L18: In result (2) the first part has no link with the second part. The idea is not coherent.
L30: “in cold areas does not fit into the phrase”
L32: “our field trip” this is not clear
L41: please replace “believe” with “found” or “discovered”. The science is not a question of to believe, but to demonstrate or to discover.
L43-44: “significantly increased and then declined”: this is definitely not clear
L46: double point
L52: the acronym for MI has been already given
L93: In scientific studies we should avoid affective consideration as “our country”. This could be simply replaced with China.
L109: Is not clear what 1a does represents.
L112: I think that is pro-duct
L120: parameters instead of elements
L124-126: Please rephrase. This is hardly understandable.
L142: Palmer instead of palmer
L204: I think you refer here on distribution, not on pattern.
L205-206: Why the full period is fragmented?
L264: Random comma in the middle of the line
L418-419: The first sentence is not coherent.
Graphical aspects:
Figure 2 should indicate subplots a) and b). Actually, I don’t understand how the blue line trend was constructed based only in two points.
Figure 8-9 should indicate also subplots. Also, for all the figures in the manuscript, the title captions should be detailed with all the elements for a full description of the displayed information. As well, each subplot needs to be referenced in the text of the manuscript. On the top figure, the cone of significance needs to be indicated.
The manuscript entitled “Remote sensing based research…” analysis the climatic conditions related to water availability at ground level (precipitation, evaporation) in Heilongjiang region of China. The study presents interest for the readers by its vast analyzed region located in cold climatic conditions, but also by the complex methodological approach on which various statistical methods are deployed. Despite these general strong points, the study presents some major weaknesses that should lead to a thorough revision of the manuscript if they will be properly addressed. Among these I can mention the elementary description of the results, the poor synthesis, the fragile discussion of the results and the visible flaw in the interpretation of the obtained results. My comments on the manuscript have been sent to the authors. In conclusion, my decision suggests a major revision of the manuscript, but Rejection in order to have sufficient time for a further resubmission could be also taken to account.
Round 2
Reviewer 2 Report
The authors struggled to respond to most of my comments and I consider that the manuscript is closer to acceptance. I still have a minor observation regarding one of their response on point 10 that is not logical. Water vapor cannot significantly reduce sunshine duration since the water vapor are transparent to sunlight. They can indeed absorb heat from solar radiation, but not the visible wavelengths that determines sunshine. Please be more careful in this elementary aspect. Thus, the incoherence observed in point 10 should still be explained.
